# Severity and Management of Adverse Drug Reactions Reported by Patients and Healthcare Professionals: A Cross-Sectional Survey

**DOI:** 10.3390/ijerph20043725

**Published:** 2023-02-20

**Authors:** Warisara Srisuriyachanchai, Anthony R. Cox, Sirinya Kampichit, Narumol Jarernsiripornkul

**Affiliations:** 1Division of Clinical Pharmacy, Faculty of Pharmaceutical Sciences, Khon Kaen University, Khon Kaen 40002, Thailand; 2School of Pharmacy, Institute of Clinical Sciences, College of Medical and Dental Sciences, University of Birmingham, Birmingham B15 2TT, UK; 3Department of Pharmacy Service, Srinagarind Hospital, Faculty of Medicine, Khon Kaen University, Khon Kaen 40002, Thailand

**Keywords:** patient self-reporting, adverse drug reactions, severity, management, prevention

## Abstract

Adverse drug reaction (ADR) severity levels are mainly rated by healthcare professionals (HCPs), but patient ratings are limited. This study aimed to compare patient-rated and pharmacist-rated ADR severity levels and determined methods employed for ADR management and prevention by patients and HCPs. A cross-sectional survey was conducted in outpatients visiting two hospitals. Patients were asked about ADR experiences using a self-administered questionnaire, and additional information was retrieved from the medical records. In total, 617 out of 5594 patients had experienced ADRs (11.0%), but 419 patients were valid (68.0%). Patients commonly reported that their ADR severity level was moderate (39.4%), whereas pharmacists rated the ADRs as mild (52.5%). There was little agreement between patient-rated and pharmacist-rated ADR severity levels (κ = 0.144; *p* < 0.001). The major method of ADR management by physicians was drug withdrawal (84.7%), while for patients, it was physician consultation (67.5%). The main methods for ADR prevention by patients and HCPs were carrying an allergy card (37.2%) and recording drug allergy history (51.1%), respectively. A higher level of ADR bothersomeness was associated with higher ADR severity levels (*p* < 0.001). Patients and HCPs rated ADR severity and used ADR management and prevention methods differently. However, patient rating of ADR severity is a potential signal for severe ADR detection of HCPs.

## 1. Introduction

Adverse drug reactions (ADRs) are a leading cause of morbidity and mortality [1], and it has been estimated that ADRs are the fourth to sixth highest cause of death in the USA [2]. However, ADR reporting rates are low, with estimated reporting rates between 9 and 15% [2,3,4,5,6]. In one study, the overall incidence of ADRs, including non-serious and serious ADRs, was found to be 10.9% in hospital patients, with 2.1% of ADRs rated as serious and 0.19% as fatal [2]. Although the majority of detected ADRs are mild to moderate [7,8,9,10,11,12], it is the severe ADRs that significantly increase health expenditure [12]. The main independent factors associated with increased severity of ADRs are advanced age, female sex, and concomitant drug use [8,12,13,14,15].

Spontaneous reporting systems (SRSs) are the most commonly employed method of pharmacovigilance utilized by healthcare professionals (HCPs) to report ADRs worldwide. The major limitation of spontaneous reports is under-reporting, with a systematic review estimating that 6% of ADRs are reported [16]. The introduction of patient self-reporting of ADRs has increased reporting rates and is now accepted in many countries [17,18,19,20]. One study found that patient self-reports had the advantage of identifying possible new ADRs that had not previously been reported by HCPs and that the quality of patient reports was equivalent to those provided by HCPs [21]. Several studies have claimed that patients report more extensive detail than reports from HCPs and describe the impact of ADRs on their lives more clearly [18,19,22].

In Thailand, both HCPs and patients can report suspected ADRs to the Health Product Vigilance Center (HPVC), but it is pharmacists that have the main responsibility for ADR monitoring and reporting. Since 2010, patients have been allowed to notify the ADRs to the HPVC. The HPVC has emphasized the importance of ADR monitoring and reporting as a standard drug surveillance method for hospital pharmacists, and hospital pharmacists play an especially important role in the detection, identification, management, and prevention of serious ADRs [23]. The study hospitals mainly used SRS for ADR monitoring, which are regulated by the HPVC as standard drug surveillance methods for hospital pharmacists nationwide.

The severity of ADRs can be measured through standard criteria such as Hartwig’s Severity Assessment Scale, which focuses on objective data and clinically relevant information. In several studies, the incidence of severe ADRs reported by HCPs using the Hartwig Scale ranged from 2% to 10% [9,10,11,12] in comparison to a rate of 20% severe ADRs reported by patients using a visual analog scale (VAS) [8]. The higher incidence of severe ADRs reported by patients reflects the more subjective impact of ADRs on their lives. Most previous studies have rated the severity levels of ADRs by HCPs [9,10,11,12,24] with comparatively limited analysis of patient-rated ADR severity levels [8,25,26,27].

The major ADR management method used by patients [28,29] and HCPs [12,13] is to withdraw the drug suspected of causing the ADR. Other methods are dose management or the use of alternative drugs [30]. HCPs may also use computerized methods to monitor treatment [31] or provide an ADR sticker on the patient tag to manage ADRs [32]. Little is known regarding the comparison between patient- and pharmacist-rated severity levels of ADRs. Therefore, the current study explored patient ADR experiences in order to compare the patient- and pharmacist-rated severities of ADRs, as well as to evaluate ADR management and prevention methods used by patients and HCPs.

## 2. Materials and Methods

### 2.1. Study Design and Setting

A cross-sectional study was conducted at outpatient departments of two university hospitals in Northeastern Thailand, namely, Srinagarind Hospital and Queen Sirikit Heart Center of the Northeast, during January to July 2020. Srinagarind Hospital is a teaching hospital, and the Queen Sirikit Heart Center is a tertiary hospital, and both are in the North East of Thailand. Patients who visited both study hospitals were those referred from all community hospitals and general hospitals in the North East region of Thailand. A retrospective study was conducted by exploring the ADRs that patients experienced. Additional information was retrieved from the medical records database including suspected drug, concomitant drugs, patient disease states, signs and symptoms, physical examination, laboratory data, doctors’ and nurses’ notes, and management and prevention of ADRs recorded by HCPs.

### 2.2. Participants

Eligible participants were patients aged 18 years and over who had most recently experienced ADRs in the past. The study excluded patients who could not read or communicate in the Thai language and patients whose information could not be obtained from the medical records. Sample size was calculated using the Cochrane equation [33] with a 5% margin of error, and 22.4% of the patients reported severe ADRs, with the rate of refusal to participate being similar to that in a previous study at 20% [8,28]. The total number of participants required was at least 400.

### 2.3. Questionnaire Development

The self-administered questionnaire was developed from our previous studies [27,34,35]. The questionnaire consisted of two sections.

Section 1 consisted of closed questions with a checklist to obtain demographic data. Sex, education level, occupation, income, and information on underlying chronic diseases were collected. An open question was used to obtain age.

Section 2 examined the management and prevention of ADRs and the ADR monitoring process by patients and HCPs. Open-ended questions were used to obtain the name of suspected drugs, indications, and the ADR symptom respondents were most confident in identifying. Participants were also asked to evaluate the severity level of ADRs using a Numerical Rating Scale (NRS) (range 0–10), scored from 0 (no severe) to 10 (most severe). Patients marked “X” on the number 0–10 in NRS line according to their assessment. Closed questions with a checklist were used to obtain information on ADR management and prevention by patients.

### 2.4. Questionnaire Testing

Three HCPs with proficiency in the field of ADRs validated the content of the developed questionnaire by examining the appropriateness of all questions in accordance with study objectives. The three HCPs included one physician who was regularly involved in ADR monitoring and reporting in Khon Kaen Hospital, as well as two clinical pharmacists who were responsible for ADR monitoring at the study hospitals.

The index of consistency (IOC) was calculated to assess validity. All questions passed content validity testing with IOC > 0.5 for each item. The questionnaire was thereafter adjusted and piloted in 15 patients at outpatient clinics in the study hospitals to evaluate patients’ ability to complete the questionnaire and to obtain any suggestions for improving the questionnaire. The piloted patients were excluded from the main study and final analysis. The final questionnaire was re-adjusted following recommendations from the pilot study and validation testing.

### 2.5. Questionnaire Distribution and Data Collection

#### 2.5.1. Patient Experience

The final version of the self-administered questionnaire was directly distributed to 419 outpatients by purposive sampling to include patients having experienced ADRs. The participants were over 18 and had the most recently experienced ADRs in the past. The symptoms that patients were most confident in identifying as an ADR were rated their ADR severity levels. The participants were able to complete the questionnaire themselves. The researcher provided assistance by reading the questionnaire to respondents with visual problems but without providing further explanation. The outpatients were recruited while they were waiting to meet physicians at outpatient clinics or to receive the drugs at pharmacy department at two university hospitals. One research pharmacist was responsible for collecting all the patient experience of ADRs in the past to minimize bias of data collection process. Consent forms were given to ask patients’ permission to participate in the study.

#### 2.5.2. Additional Information from Medical Records

Additional information about outpatients reporting ADRs was retrieved from the medical records database. The retrieved data consisted of general information, drug therapy, ADR information, and ADR management and prevention by HCPs. The Hartwig Scale was used to assess the severity of ADRs by a research pharmacist. The Hartwig Scale categorizes ADRs into seven levels of severity. Levels 1 and 2 are mild, levels 3 and 4 are moderate, and levels 5, 6 and 7 are severe. Mild ADRs were defined as self-limiting and able to resolve over time without treatment. Moderate ADRs were defined as those that required treatment or increased length of stay by at least one day. Severe ADRs were defined as those that were life-threatening or caused permanent harm or death [36]. Patients assessed the severity of their suspected ADRs by the ten-point NRS. Patient-reported severity levels of ADR symptoms were classified as mild (score 0–3), moderate (score 4–6), or severe (score 7–10) [37]. Naranjo’s algorithm [38] and the World Health Organization Uppsala Monitoring Centre (WHO-UMC) criteria [39] were subsequently used by a pharmacist for causality assessment of the ADRs patients were most confident in reporting. The Naranjo algorithm has four categories: definite (score > 9), probable (score 5–8), possible (score 1–4), and doubtful (score < 0) [38]. The four WHO criteria categories are certain, probably/likely, possible, and unlikely [39].

### 2.6. Data Analysis

The questionnaires were analyzed using SPSS for Windows version 26.0. Demographic data, frequencies, and types of ADR were analyzed using descriptive statistics. Pearson’s chi-squared test and Fisher’s exact test were used to compare subgroups for categorical data, and independent-sample *t*-tests were used for continuous data. The Mann–Whitney U test was used to determine the independence of non-parametric variables. The patient-reported severity level of ADR symptoms were represented in a score range of 0–10 and are presented with either mean ± SD or median and range. The levels of severity of ADRs were divided into binary variables as follows: mild and moderate to severe. Agreement between NRS and the Hartwig Scale was judged to be acceptable on the basis of a Cohen’s kappa (κ). Cohen’s kappa (κ) was interpreted as follows: poor < 0, slight = 0.01–0.20, fair = 0.21–0.40, moderate = 0.41–0.60, substantial = 0.61–0.80, and almost perfect agreement = 0.81–1.00 [40]. Univariate analysis of the factors related to the variables was conducted using Pearson’s chi-squared test. Variables associated with severe ADRs at *p*-values ≤ 0.25 in the univariate analyses were entered into a multivariate analysis. Multivariate analysis of the factors that related to severity level of ADR symptoms reported by patients was conducted using logistic regression. Results with *p*-values less than 0.05 were considered statistically significant. Potential sources of bias might be recall bias, selection bias, and under-representation of populations. As underlying diseases and advanced age might be confounding variables [7,41], these variables were subgrouped for data analysis.

## 3. Results

### 3.1. Response Rate

A total of 5594 patients were approached and asked about their experience of suspected medicine-related symptoms, and 617 claimed that they had experienced ADRs (11.0%). All 617 patients were invited to participate in the study by the research pharmacist, and 419 questionnaires were completed and analyzed (response rate 68.0%). The main reasons given for refusal to participate in the study were it not being convenient (n = 63, 31.8%) and forgetting details of experienced ADRs (n = 44, 22.2%). Seven participants were excluded due to incomplete data (3.5%), 73 because drug allergy histories were not found in the medical records (36.9%), and 11 because their medical records could not be assessed (5.6%). Of the total 419 participants, 298 were collected from Srinagarind Hospital (71.1%) and 121 were from Queen Sirikit Heart Center of the Northeast, Thailand (28.9%).

### 3.2. Demographic Data

The majority of respondents were female (n = 278, 66.3%), with an average age of 59.2 ± 14.13 years (range 20–89). Half of the patients were aged more than 60 years (n = 228, 54.4%), and education levels were secondary school and lower (n = 223, 53.2%) or a bachelor’s degree and higher (n = 189, 45.1%). About one-third of respondents were employed in government or state enterprises (n = 157, 37.5%). The majority of patients had underlying diseases (n = 372, 88.8%). Half of the patients reported that their recent ADRs occurred within the last 5 years (n = 228, 54.5%). Half of the patients reported that the suspected ADRs lasted less than 24 h (n = 238, 56.7%), and almost two-thirds of respondents reported ADR duration of less than 3 days (n = 256, 61.1%) (Table 1).

Of the 419 respondents, the top three reported ADRs classified by systemic organ class (SOC) were skin tissue disorders (n = 299, 71.4%) including rash (n = 203, 48.4%), respiratory system disorders (n = 29, 6.9%) including dyspnea (n = 29, 6.9%), and gastrointestinal disorders (n = 18, 4.3%) including nausea/vomiting (n = 9, 2.1%). The most common suspected drugs classified by anatomical therapeutic chemical (ATC) with their reported ADRs were penicillin (n = 73)-induced rash, sulfamethoxazole and trimethoprim (n = 33)-induced rash, and enalapril (n = 22)-induced dry cough. The severity of the ADRs experienced by patients were reported as mild in 24.1% of cases, moderate in 39.4% of cases, and severe in 36.5% of cases. HCPs rated the severity of ADRs as mild in 52.5% of cases, moderate in 40.8% of cases, and severe in 6.7% of cases. According to the Naranjo algorithm, the number of definite ADRs was 6 (1.4%), while 155 were classified as probable (37.0%) and 257 as possible (61.3%), with 1 suspected ADR (0.2%) classified as a doubtful or false ADR. According to the WHO criteria, the number of suspected ADRs classified as certain was 9 (2.1%), with 136 probable (32.5%) and 273 possible (65.2%). One suspected ADR (0.2%) was also classified as unlikely or false according to the WHO criteria. Thus, 418 out of 419 (99.8%) of the suspected ADRs were classified as true ADRs according to both methods.

### 3.3. Agreement between Severity Assessment Tools

There was slight agreement (percentage of positive agreement = 41.1, κ = 0.144) between the NRS patient-rated and the Hartwig Scale pharmacist-rated severity levels of ADRs (Table 2).

### 3.4. ADR Management by Patients and Physicians

The top three methods of ADR management by patients were immediate physician consult (67.5%), drug withdrawal (45.4%), and continue drug in same dose (9.9%). A significant increase in the proportion of patients who reported immediate physician consult (85.4%, *p* < 0.001) was found for patients with ADRs rated as moderate to severe by the Hartwig Scale. For patients with ADRs rated as moderate to severe by NRS, there was a significant increase in the proportion of patients reporting immediate physician consult (71.5%, *p* = 0.002) (Table 3).

The major method of ADR management by physicians was drug withdrawal (84.7%). For patients who rated the severity level of ADRs by NRS, there were no significant differences for all methods of ADR management. Patients with ADRs rated as moderate to severe by the Hartwig Scale were found to have a higher proportion managed by drug withdrawal (90.4%), followed by giving another drug for treating ADRs (53.4%). A significant difference in the physicians’ ADR management method was only seen for giving another drug for treating ADRs (*p* < 0.001) (Table 3).

### 3.5. ADR Prevention by Patients and HCPs

The carrying of allergy cards was the only method that patients used for prevention of their ADRs (n = 156, 37.2%). Regarding ADR prevention by HCPs, 214 (51.1%) patient ADRs were recorded in the medical records or electronic records, 173 (41.3%) patients had been given an allergy card, but only 55 (13.1%) had a sticker attached to their medical records. Patients with ADRs rated as moderate to severe by NRS were found to have a higher proportion that received an allergy card (n = 140, 44.0%) and attached a sticker to medical records (n = 43, 13.5%)), but a significant difference was only seen for received allergy cards (*p* = 0.044). Patients with ADRs rated as moderate to severe by the Hartwig Scale were found to have a higher proportion of all methods of ADR prevention, and significant differences were seen in the medical records or electronic records (*p* = 0.003), received allergy cards (*p* < 0.001), and attached stickers to medical records (*p* = 0.022) (Table 4).

### 3.6. Reported ADRs and Suspected Drugs in Patients with Severe ADRs (Rated by NRS)

The most common reported ADRs that resulted in immediate physician consultations by patients and drug withdrawal by HCPs were rash, dyspnea, and angioedema, which were similar for all methods of ADR prevention by patients and HCPs. For ADRs that resulted in a decrease in the dose by HCPs, the most commonly reported ADRs were dependent on the pharmacological effect of the drug such as ankle edema or bleeding. The main suspected drugs that were identified by most methods of ADR management and prevention were antibiotics and NSAIDs. ADRs resulting from cardiovascular drugs such as enalapril, amlodipine, and statins were managed by decreasing dose or changing to an alternative drug (Table 5).

### 3.7. Factors Related to Severity Levels of ADR Symptoms

The univariate analysis of the factors affecting the severity levels of ADRs that were rated by NRS showed that sex (*p* < 0.001), age (*p* = 0.085), and increased levels of ADR bothersomeness (*p* < 0.001) were significantly associated with higher levels of ADR severity. The follow-up multiple logistic regression analysis showed that the factors independently associated with severity levels of ADRs were a medium level of ADR bothersomeness (OR 3.678; 95% CI 2.029, 6.666; *p* < 0.001) and a high level of ADR bothersomeness (OR 26.211; 95% CI 9.077, 75.658; *p* < 0.001) (Table 6).

The univariate analysis of the factors affecting the severity levels of ADRs that were rated by the Hartwig’s Severity Assessment Scale showed that sex (*p* = 0.063), education level (*p* = 0.166), underlying diseases (*p* = 0.033), onset of ADRs (*p* = 0.092), duration of ADRs (*p* = 0.155), and increased levels of ADR bothersomeness (*p* < 0.001) were significantly associated with levels of ADR severity. The follow-up multiple logistic regression analysis showed that the factors independently associated with higher severity levels of ADRs were medium level of ADR bothersomeness (OR 2.300; 95% CI 1.359, 3.892; *p* = 0.002) and high level of ADR bothersomeness (OR 3.951; 95% CI 2.372, 6.581; *p* < 0.001). The factors independently associated with lower severity levels of ADRs were duration of ADRs between 4 day and 4 weeks (OR 0.441; 95% CI 0.256, 0.758; *p* = 0.003) and duration of ADRs more than 1 month (OR 0.388; 95% CI 0.180, 0.834; *p* = 0.015) (Table 6).

## 4. Discussion

The percentages of male and female individuals participating in this study were 33.7% and 66.3%, respectively, which is similar to similar studies undertaken in Thai populations [8,35]. Moreover, in line with previous studies, the most frequently reported ADRs as classified by systemic organ class (SOC) were skin tissue disorders [28,35,42], and the most common drug group suspected of causing ADRs was anti-infectives [12,28,42,43]. Previous studies have provided limited data to compare severity rating scales between patients and HCPs [8,9,10,11,12,24,25,26]. The major strength of our study is the rating of ADR severity levels by both patients and HCPs. Patients’ ADR reports have been shown to be more extensive in the amount of detail provided about reactions and more often describe the effects of ADRs on their’ lives [22]. On the other hand, ADR reports from HCPs are considered to provide more objective data and are based on clinically related information [19].

Patients usually report possible connections between suspected drugs and their symptoms, and they tend to report common, less-serious symptoms that affect their daily life [7,8,22,27,28,44]. In contrast, HCPs often report more serious ADRs [45]. This study showed that patients frequently rated ADRs as moderate to severe, whereas HCPs frequently rated ADRs as mild. A previous study found that a low number of reported symptoms and a high level of confidence were two factors that were related to accuracy in ADR identification [35]. Moreover, a temporal relationship between the onset of symptoms and the taking of a drug was the main issue that patients used to confirm a suspected ADR [7,18]. This study showed that the NRS patient-rated and the Hartwig Scale pharmacist-rated severity levels of ADRs were not a perfect match. NRS patient ratings provided a high proportion of moderate-to-severe ADRs, but the Hartwig Scale pharmacist ratings provided high proportion of mild-to-moderate ADRs. Patient ratings showed a slight but significant agreement (κ = 0.144) with the Hartwig Scale pharmacist ratings. These findings indicate that patients and pharmacists rate the severity levels of ADRs differently. Nevertheless, patients should still notify the ADR severity to pharmacists for further assessment.

Interestingly, some patients (12.6%) rated ADR severity level as severe, but HCPs rated it as mild, which may affect the patients’ adherence to treatment [46]. On the other hand, for ADRs that patients rated as mild, this may lead to patients not reporting their ADR symptoms to HCPs, with potentially serious consequences of severe ADRs. Increasing patient awareness of ADR monitoring, including paying attention to ADR information or ADR bothersomeness, may reduce the chance of harm for severe ADRs. Improving patients’ awareness in the observation of ADR symptoms or laboratory data, ADR knowledge and increase communication to HCPs should be promoted. Despite differences in ADR severity level assessments by patients and HCPs, there are potential benefits from sharing of both assessments. HCPs tend to assess the severity level of ADRs on the basis of objective data and clinically relevant information, whereas patients assess the severity of ADRs from subjective data about how the symptoms affect their daily life [22]. Sharing assessments of ADR severity levels between patients and HCPs could have potential benefits for early detection of severe ADRs.

The current study found that the main method patients used for management of ADRs was to directly consult with a physician, but previous studies have shown that patients stop taking their medication by themselves after experiencing an ADR [28,29]. The most frequent ADR management method applied by HCPs was drug withdrawal [12,13,47], which is a method considered suitable to manage severe ADRs [47]. Other methods used by HCPs such as dose reductions were used in response to ADRs resulting from the drugs pharmacological action. The most frequent ADR management in non-serious ADRs were pharmacologic and non-pharmacologic treatment or dose management [30]. The ADR severity levels were related to the different methods used for ADR management.

The most frequent ADR prevention method reported by HCPs was noting the ADR in the medical records or electronic health records, in line with other studies using computer alerts [31,48]. Improvements in hospital database systems and providing support to physicians to use them may have a beneficial impact on the safety of medical treatment [48]. In addition to annotating electronic health records, providing an allergy card or attaching a sticker to the medical records were the major methods used to prevent and minimize ADRs. The major ADR prevention method reported by patients was carrying an allergy card, one of the most effective methods used in one study [32].

In this study, univariate analysis showed that sex, age, and level of ADR bothersomeness were the factors related to severity level of ADRs (as rated by NRS). Female patients seem to be at greater risk of developing severe ADRs [12,49], as are patients of advanced age [8,12]. Patients aged 45–60 years have a 1.97-fold greater risk of developing severe ADRs compared to the rest of the population, and a previous study showed that patients aged 30–50 years experience more severe ADRs [8]. However, the multivariate analysis found that it was only higher levels of ADR bothersomeness that were associated with ADR severity, which is similar to the findings of a previous study [8]. Moreover, univariate analysis showed that sex, duration of ADR symptoms, and level of ADR bothersomeness were the factors related to severity level of ADRs (rated by the Hartwig’s Severity Assessment Scale). However, ADR severity levels or seriousness was evaluated by physicians on the basis of life threatening patterns of ADRs.

Currently, information on patient-rated and pharmacist-rated severity of ADRs has been limited. The strengths of the study were that this is the first study to compare patient-rated and pharmacist-rated severity levels of ADRs and the pharmacist-rated ADR severity levels made by one research pharmacist using standard criteria to minimize the bias of the assessment. The current study has some limitations. Incomplete data retrieving from the medical records database led to a number of invalid participants for data analysis. However, the study was conducted in the northeastern region of Thailand, and hence the findings may not be generalized to all patients and HCPs in Thailand. Moreover, the gathered findings were obtained from a self-administered questionnaire, which may be subject to recall and social desirability biases.

## 5. Conclusions

Patients and HCP ratings of ADR severity differ with practical consequences for reporting and management of ADRs. Patients were more likely to rate ADR severity as severe ADRs due to the effects on their daily life, but HCPs tend to assess ADR severity levels using more clinically relevant information. Higher levels of ADR bothersomeness were significantly associated with severe ADRs when rated by patients and HCPs. Methods of ADR management and prevention varied between severity levels. However, patient self-rating of ADR severity may support ADR monitoring by HCPs. Patient- and HCP-rated ADR severity level should be implemented in practices.

## Figures and Tables

**Table 1 ijerph-20-03725-t001:** Demographic characteristics of respondents.

Characteristics	Number of Patients; N (%)			*p*-Value ^a^
	Mild ^c^ (n = 101)	Moderate to Severe ^c^ (n = 318)	Total (n = 419)	
Sex				
Male	51 (50.5)	90 (28.3)	141 (33.7)	<0.001 *
Female	50 (49.5)	228 (71.7)	278 (66.3)	
Age (years)				
18–44	15 (14.9)	51 (16.0)	66 (15.8)	0.085
45–60	22 (21.8)	103 (32.4)	125 (29.8)	
>60	64 (63.4)	164 (51.6)	228 (54.4)	
Mean ± SD	60.0 ± 13.36	58.9 ± 14.37	59.2 ± 14.13	0.434 ^b^
Median (range)	64 (24–85)	61 (20–89)	62 (20–89)	
Occupation				
No carrer	23 (22.8)	68 (21.4)	91 (21.7)	0.397
Farmer + worker	20 (19.8)	89 (28.0)	109 (26.0)	
Government + state enterprises	43 (42.6)	114 (35.8)	157 (37.5)	
Own bussiness	15 (14.9)	47 (14.8)	62 (14.8)	
Education level				
Secondary school and lower	52 (51.5)	171 (53.8)	223 (53.2)	0.540
Bachelor’s degree and higher	49 (48.5)	140 (44.0)	189 (45.1)	
Unknown			7 (1.7)	
Underlying diseases				
Yes	92 (91.1)	280 (88.1)	372 (88.8)	0.484
No	9 (8.9)	36 (11.3)	45 (10.7)	
Unknown			2 (0.5)	
When most recent ADRs occurred				0.101
Within the past week	7 (6.9)	15 (4.7)	22 (5.3)	
1–4 weeks ago	4 (4.0)	12 (3.8)	16 (3.8)	
1–6 months ago	16 (15.8)	27 (8.5)	43 (10.3)	
6–12 months ago	5 (5.0)	19 (6.0)	24 (5.7)	
1–5 years ago	20 (19.8)	103 (32.4)	123 (29.3)	
More than 5 years ago	47 (46.5)	140 (44.0)	187 (44.6)	
Unknown			4 (1.0)	
Onset of ADRs				
Acute (<1 h)	21 (20.8)	81 (25.5)	102 (24.3)	0.630
Subacute (1–24 h)	35 (34.7)	101 (31.8)	136 (32.4)	
Latent (>24 h)	44 (43.6)	134 (42.1)	178 (42.5)	
Unknown			3 (0.7)	
Duration of ADRs				
<1–24 h	37 (36.6)	107 (33.6)	144 (34.4)	0.719
1–3 days	27 (26.7)	85 (26.7)	112 (26.7)	
4 days–4 weeks	22 (21.8)	87 (27.4)	109 (26.0)	
>1 month	8 (7.9)	32 (10.1)	40 (9.5)	
Unknown			14 (3.3)	
Bothersome ADRs				
Low	57 (56.4)	85 (26.7)	142 (33.9)	<0.001 *
Medium	13 (12.9)	86 (27.0)	99 (23.6)	
High	4 (4.0)	132 (41.5)	136 (32.5)	

^a^ Pearson’s chi-squared test, ^b^ Mann–Whitney U test, ^c^ ADR severity level were rated by the Numerical Rating Scale (NRS), * the level of significant difference <0.05.

**Table 2 ijerph-20-03725-t002:** Agreement of severity assessment between the Numerical Rating Scale and Hartwig’s Severity Assessment Scale (n = 419).

ADR Severity Level		Numerical Rating Scale			
		Mild	Moderate	Severe	Total
Hartwig’s Severity Assessment Scale	Mild	75 (34.1)	92 (41.8)	53 (2.41)	220 (100.0)
	Moderate	26 (15.2)	71 (41.5)	74 (43.3)	171 (100.0)
	Severe	0 (0.0)	2 (7.1)	26 (92.9)	28 (100.0)
	Total	101 (24.1)	165 (39.4)	153 (36.5)	419 (100.0)
Percentage of positive agreement		172 (41.1)			
Kappa		0.144			
*p*-value		<0.001			

**Table 3 ijerph-20-03725-t003:** Relationship between management of ADRs by patients and HCPs in relation to severity level by the Numerical Rating Scale and Hartwig’s Severity Assessment Scale.

**Methods**	**Management by Patients** **; N (%)**							
	**NRS ^a^ by Patients**				**Hartwig ^b^ by Pharmacist**			
	**Mild** **(** **n = 100)**	**Moderate to Severe** **(** **n = 316)**	**Total ^#^** **(** **n = 416)**	** *p* ** **-Value ^c^**	**Mild** **(** **n = 218)**	**Moderate to Severe** **(** **n = 198)**	**Total** **(** **n** **=** **416)**	** *p* ** **-Value ^c^**
Drug withdrawal	48 (48.0)	141 (44.6)	189 (45.4)	0.554	117 (53.7)	72 (36.4)	189 (45.4)	<0.001 *
Continue drug (same dose)	11 (11.0)	30 (9.5)	41 (9.9)	0.660	27 (12.4)	14 (7.1)	41 (9.9)	0.069
Use another drug for treating ADRs	1 (1.0)	3 (0.9)	4 (1.0)	1.000 ^d^	2 (0.9)	2 (1.0)	4 (1.0)	1.000 ^d^
Consult with community pharmacist	8 (8.0)	8 (2.5)	16 (3.8)	0.019 ^d^	9 (4.1)	7 (3.5)	16 (3.8)	0.753
Searching for more information	6 (6.0)	10 (3.2)	16 (3.8)	0.231 ^d^	13 (6.0)	3 (1.5)	16 (3.8)	0.018 *
Immediate physician consultation	55 (55.0)	226 (71.5)	281 (67.5)	0.002 *	112 (51.4)	169 (85.4)	281 (67.5)	<0.001 *
**Method** **s**	**Management by Physicians** **; N (%)**							
	**NRS ^a^ by Patients**				**Hartwig ^b^ by Pharmacist**			
	**Mild** **(** **n = 34)**	**Moderate to Severe** **(** **n = 110)**	**Total ^$^** **(** **n = 144** **)**	** *p* ** **-Value ^c^**	**Mild** **(** **n = 71)**	**Moderate to Severe** **(** **n** **=** **73)**	**Total** **(** **n = 144)**	** *p* ** **-Value ^c^**
Drug withdrawal	29 (85.3)	93 (84.5)	122 (84.7)	0.916	56 (78.9)	66 (90.4)	122 (84.7)	0.054
Continue drug (decrease dose)	3 (8.8)	12 (10.9)	15 (10.4)	1.000 ^d^	11 (15.5)	4 (5.5)	15 (10.4)	0.049 *
Continue drug (same dose)	2 (5.9)	7 (6.4)	9 (6.3)	1.000 ^d^	6 (8.5)	3 (4.1)	9 (6.3)	0.323 ^d^
Change to alternative drug	12 (35.3)	40 (36.4)	52 (36.1)	0.910	37 (52.1)	15 (20.5)	52 (36.1)	<0.001 *
Giving another drug for treating ADRs	11 (32.4)	30 (27.3)	41 (28.5)	0.566	2 (2.8)	39 (53.4)	41 (28.5)	<0.001 *

^a^ NRS: Numerical Rating Scale: mild (0–3), moderate (4–6), and severe (7–10). ^b^ Hartwig: Hartwig’s Severity Assessment Scale: mild (1–2), moderate (3–4), and severe (5–7). ^c^ Pearson’s chi-squared test, ^d^ Fisher’s exact test, * the level of significant difference < 0.05. ^#^ Patients did not specify in the questionnaire (n = 3). ^$^ No data found in medical records (n = 275).

**Table 4 ijerph-20-03725-t004:** Relationship between prevention of ADRs by HCPs in relation to severity level by the Numerical Rating Scale and Hartwig’s Severity Assessment Scale.

Methods	Prevention by HCPs; N (%)							
	NRS ^a^ by Patients				Hartwig ^b^ by Pharmacist			
	Mild (n = 101)	Moderate to Severe (n = 318)	Total (n = 419)	*p*-Value ^c^	Mild (n = 220)	Moderate to Severe (n = 199)	Total (n = 419)	*p*-Value ^c^
Medical record/ computer popup	43 (42.6)	171 (53.8)	214 (51.1)	0.055	97 (44.1)	117 (58.8)	214 (51.1)	0.003 *
Give allergy card	33 (32.7)	140 (44.0)	173 (41.3)	0.044 *	66 (30.0)	107 (53.8)	173 (41.3)	<0.001 *
Attach sticker to medical record	12 (11.9)	43 (13.5)	55 (13.1)	0.671	21 (9.5)	34 (17.1)	55 (13.1)	0.022 *

^a^ NRS: Numerical Rating Scale: mild (0–3), moderate (4–6), and severe (7–10). ^b^ Hartwig: Hartwig’s Severity Assessment Scale: mild (1–2), moderate (3–4), and severe (5–7). ^c^ Pearson’s chi-squared test, * the level of significant difference <0.05.

**Table 5 ijerph-20-03725-t005:** Top three most commonly reported ADRs and suspected drugs in patients with severe ADRs (rated by NRS ^a^) in relation to management and prevention by patients and HCPs.

Methods	% of Patients	
	ADR Symptoms	Suspected Dugs
**Management of ADRs by patients** (n = 153)		
Drug withdrawal (n = 60, 39.2%)	Rash (46.7%), angioedema (10.0%), dyspnea (5.0%)	Penicillin (15.0%), cotrimoxazole (8.3%), diclofenac (6.7%)
Continue drug (same dose) (n = 16, 10.5%)	Rash (25.0%), dry cough (18.8%), nausea (12.5%)	Enalapril (18.8%), amoxicillin and clavulanic acid (6.3%), amlodipine (6.3%)
Use another drug for treating ADRs (n = 1, 0.7%)	Ankle edema (100.0%)	Amlodipine (100.0%)
Consult community pharmacist (n = 4, 2.6%)	Rash (75.0%), nausea (25.0%)	Amoxicillin and clavulanic acid (25.0%), amoxicillin (25.0%), penicillin (25.0%)
Searching for more information (n = 7, 4.6%)	Angioedema (14.3), dry cough (14.3%), dyspnea (14.3%)	Carboplatin (14.3%), diclofenac (14.3%), enalapril (14.3%)
Immediate physician consult (n = 112, 73.2%)	Rash (53.6%), angioedema (8.0%), dyspnea (8.0%)	Penicillin (15.2%), cotrimoxazole (7.1%), amoxicillin (5.4%)
**Management of ADRs by HCPs** (n = 153)		
Drug withdrawal (n = 53, 34.6%)	Rash (45.3%), dry cough (11.3%), myositis (9.4%)	Enalapril (11.3%, cotrimoxazole (7.5%), atorvastatin (5.7%)
Continue drug (decrease dose) (n = 4, 2.6)	Ankle edema (25.0%), face edema (25.0%), bleeding (25.0%)	Amlodipine (25.0%), prednisolone (25.0%), warfarin (25.0%)
Continue drug (same dose) (n = 5, 3.3%)	Headache (20.0%), hoarseness (20.0%), mucositis (20.0%)	Carboplatin (20.0%), levofloxacin (20.0%), seretide accuhaler^®^ (20.0%)
Change to alternative drug (n = 18, 11.8%)	Dry cough (33.3%), myositis (27.8%), rash (22.2%)	Enalapril (33.3%), atorvastatin (11.1%), methimazole (11.1%)
Giving another drug for treating ADRs (n = 18, 11.8%)	Rash (77.8%), angioedema (5.6%), dyspnea (5.6%)	Ultravist^®^ (16.7%), amoxicillin (11.1%), celecoxib (5.6%)
**Prevention of ADRs by patients** (n = 147)		
Carry of allergy card (n = 95, 64.6%)	Rash (55.8%), dyspnea (11.6%), angioedema (7.4%)	Penicillin (13.7%), cotrimoxazole (7.4%), amoxicillin (5.3%)
**Prevention of ADRs by HCPs** (n = 153)		
Medical record/computer popup (n = 81, 52.9%)	Rash (54.3%), dyspnea (9.9%), angioedema (8.6%)	Penicillin (13.6%), amoxicillin (6.2%), cotrimoxazole (6.2%)
Give allergy card (n = 69, 45.1%)	Rash (52.2%), dyspnea (11.6%), angioedema (5.8%)	Penicillin (10.1), diclofenac (7.2%), cotrimoxazole (5.8%)
Attach sticker to medical record (n = 25, 16.3%)	Rash (76.0%), angioedema (8.0%), dyspnea (8.0%)	Penicillin (16.0%), celecoxib (8.0%), cotrimoxazole (8.0%)

^a^ NRS: Numerical Rating Scale: mild (0–3), moderate (4–6), and severe (7–10).

**Table 6 ijerph-20-03725-t006:** Multiple logistic regression of factors related to severity levels of ADR symptoms reported by patients.

**ADR Severity Level Were Rated by Numerical Rating Scale (NRS)**						
**Variables ^a^**	**N (%)**		**Adjusted OR**	**95% CI**		** *p* ** **-** **Value**
	**Mild** **(** **n = 101)**	**Moderate to Severe** **(** **n = 318)**		**Lower**	**Upper**	
Sex						
Male	51 (50.5)	90 (28.3)	1			0.088
Female	50 (49.5)	228 (71.7)	1.614	0.932	2.795	
Age (years)						
18–44	15 (14.9)	51 (16.0)	1			
5–60	22 (21.8)	103 (32.4)	1.975	0.827	4.715	0.125
>60	64 (63.4)	164 (51.6)	1.276	0.592	2.750	0.534
Bothersome						
Low	57 (56.4)	85 (26.7)	1			
Medium	13 (12.9)	86 (27.0)	3.678	2.029	6.666	<0.001 *
High	4 (4.0)	132 (41.5)	26.211	9.077	75.683	<0.001 *
**ADR Severity Level Were Rated by the Hartwig’s Severity Assessment Scale**						
**Variable ^b^**	**N (%)**		**Adjusted OR**	**95% CI**		** *p* ** **-** **Value**
	**Mild** **(** **n = 10** **1** **)**	**Moderate to Severe** **(** **n = 318)**		**Lower**	**Upper**	
Sex						
Male	83 (37.7)	58 (29.1)	1			
Female	137 (62.3)	141 (70.9)	1.211	0.771	1.904	0.406
Age (years)						
18–44	29 (13.2)	37 (18.6)	1			
5–60	70 (31.8)	55 (27.6)	0.608	0.320	1.153	0.127
>60	121 (55.0)	107 (53.8)	0.812	0.449	1.469	0.491
Duration of ADRs						
<1–24 h	64 (30.6)	80 (40.8)	1			
1–3 day	59 (28.2)	53 (27.0)	0.603	0.357	1.020	0.059
4 day–4 weeks	63 (30.1)	46 (23.5)	0.441	0.256	0.758	0.003 *
>1 month	23 (11.0)	17 (8.7)	0.388	0.180	0.834	0.015 *
Bothersome						
Low	121 (55.0)	63 (31.7)	1			
Medium	49 (22.3)	50 (25.1)	2.300	1.359	3.892	0.002 *
High	50 (22.7)	86 (43.2)	3.951	2.372	6.581	<0.001 *

^a^ Variables included in the multiple logistic regression analysis were sex, age, and bothersome ADRs. ^b^ Variables included in the multiple logistic regression analysis were sex, age, duration of ADRs, and bothersome ADRs. * the level of significant difference <0.05.

## Data Availability

The data used in this study are available upon request from the corresponding author.

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
