# Peer review of "Severity and Management of Adverse Drug Reactions Reported by Patients and Healthcare Professionals: A Cross-Sectional Survey"

_ijerph, 2023, doi:10.3390/ijerph20043725_

Round 1

Reviewer 1 Report

The manuscript, whose main objectives were: to compare the pharmacists and patient rated severities of ADRs  and to evaluate ADR managements and prevention methods used by Health Care Professionals,  is clear and  well written.  The strength and limitations of the study are also presented.

There are few points that need to be addressed are as follows:

-       Line 194.  Authors state that the top methods for ADR management by patient were:  physician consult (67.5%),  drug withdrawal 45.4%, and no changes 9.9%.   Do some patients decided to withdraw the drug and at the same time consult the physician? Please explain why the value is higher than 100 %

-       Line 116. It is unclear whether   only one research pharmacist participated in the study. An explanation should be added in the text

-       Table 1. With respect to the duration of adverse reactions, are the authors confident that the patients could remember the days of duration of the adverse reaction? Is this information reliable?

-       Line 278.  The potential benefits of sharing ADR severity levels assessment by patients and HCPs, should be explained.

-       One of the objectives of the study was   compare the patient and pharmacist rated severities of ADR, however the discussion of these results is poor

Reviewer 2 Report

ARTICLE REVIEW: Severity and Management of Adverse Drug Reactions Reported by Patients and Healthcare Professionals

This is a cross-sectional study on the severity and management of adverse drug reactions recorded through a questionnaire by patients and by reviewing clinical records by pharmacists, and aimed to compare the levels of severity of reactions adverse reactions to drugs rated by patients versus those rated by pharmacists, and infer the methods used for the management and prevention of adverse drug reactions by patients versus health professionals. Cross-sectional survey conducted in outpatients visiting two university hospitals in northeastern Thailand, for 7 months. The pharmacists retrieved additional information from the patients' clinical records. The quality of the questionnaire data is poor, mainly due to recall bias. The choice of procedures for handling incomplete data is a complex task, since in certain situations the same procedure can produce precise estimates and in others not. Analysis of complete data must be done with caution because of the substantial loss of information that is generated. For all these reasons, it is necessary to better explain some parts of the methodology, the analysis and presentation of results could be improved, it is important that the authors add the missing data.

MAJOR SHORTCOMINGS:

1-    The title of the study (‘Severity and Management of Adverse Drug Reactions Reported by Patients and Healthcare Professionals’) doesn’t contain the design of the study (cross-sectional survey) (according to STROBE checklist).

2-    Introduction: Explain the situation in Thailand in spontaneous notification of ADRs, only HCPs can notify suspected ADRs or patients can also notify adverse effects, if yes from which date?

3-    Materials and methods:

a.    Page 2, section 2.1. Study design and setting: Explain what type of population the two hospitals serve.

b.    Page 3, section 2.4 Questionnaire testing: Specify the training of the three health professionals with competence in the field of ADR who validated the content of the development. Provide the final questionnaire in an annexe.

c.     Page 3, 2.5.1 Patient experience: Specify if the date of ADRs was recorded, that patients experienced since recall bias is frequent after six months of an adverse event.

d.    Page 3, 2.5.2 Additional information from medical records, it is said “Naranjo’s algorithm [38] and the World Health Organization 131 Uppsala Monitoring Centre (WHO-UMC) criteria [39] were subsequently used by a pharmacist for causality assessment of the ADRs patients were most confident in reporting.” It remains to be explained from what score of each algorithm each adverse effect will be considered as an adverse drug reaction (at least one related drug).

e.    Page 3, section 2.6 Data analysis: Potential sources of bias are not specified. Potential confounding, or modifying effect variables are not mentioned.

4-    Results:

a.    Provide the result of the causality assessment that classified as true ADRs (at least one related drug) the adverse effect recorded int the questionnaire, ), according to the Naranjo algorithm and according to the WHO criteria.

b.    Add "unknown" on each item when Incomplete data retrieving, section by section of results and table by table.

c.     Add an agreement analysis (kappa) of management and of prevention between patients and HCPs.

d.    Page 9, Table 6 (‘Multiple logistic regression of factors related to severity levels of ADR symptoms reported by patients’): multiple logistic regression has only been conducted based on ADR severity level rated by numerical rating scale (NRS) by patients. I suggest to add a multiple logistic regression based on Hartwig’s Severity Assessment Scale, and compare the results.

e.    Results of ADRs notified to a pharmacovigilance database by HCPs and by patients if applicable.

5-    Discussion:

a.    Discuss the agreement in management and of prevention of ADRs by patients with respect to HCPs.

b.    Summarize and discuss the type of adverse effect reported by patients and health care providers and the factors that affect patient reports of adverse drug reactions, the authors have much of their own data from previous studies as well as from the international bibliography.

MINOR SHORTCOMINGS:

1-    Results: Page 7, Table 6 p-value 0.055* (Medical record/ Computer popup)  it is not significant (<0.05).

2-    Results: Page 8, Table 5 (‘Top three most commonly reported ADRs and suspected drugs in patients with severe ADRs (rated by NRS) in relation to management and prevention by patients and HCPs’), in section ‘Management of ADRs by HCPs (n=153)’, specifically in the first line ‘Drug withdrawal (n=53, 34.6%)’, second column (‘ADR symptoms’): in ‘dry cough (113%)’, a dot is lacking, it should be ‘11.3%’ instead of ‘113%’.

3-    Page 10, line 289 the phase “The ADR severity levels related to the used methods of ADR management” seems incomplete.

4-    References:

a.    In references 1 and 39: links doesn’t work. Moreover, the information ‘(accessed on (Date))’ should be placed between brackets, not parentheses, and should be located before the reference link, and followed by a dot. What is more, brackets containing ‘Internet’ should be written after the website institution or organisation.

b.    In references 2 to 47:

                                          i.    Authors’ name format is not written following Vancouver norms. It should be written as: Surname (followed by a space, not a comma), then the initial letter of the name, in capitals, not followed by a dot. Between authors’ names, a comma should be placed instead of a semicolon.

                                         ii.    References with more than 6 authors: ‘et al’ should be written after mentioning the first 6 authors, according to Vancouver norms.

c.     In references 2 to 38 and 40 to 47: according to Vancouver norms, year of publication should be separated from the name of the journal by a dot. Also, between year of publication and volume of journal a semicolon should be written, not a comma. Year of publication shouldn’t be written in bold. What is more, between volume(issue) and number of pages, a colon should be placed instead of a comma. Finally, number of pages and doi should be separated by a dot.

d.    I couldn’t find reference 11 in English on Google Scholar or on PubMed.

e.    I couldn’t find reference 41 on Google Scholar or on PubMed.

Regards,

Reviewer 3 Report

1. Explain the inclusion and exclusion criteria in more elaborate on validating 419 selection.

2. Please explain the pharmacist involvement and reporting ADR.

3. Role of the Clinical Pharmacist can be well noted, and clarity can be included in the study.

4. Explain the ADR reporting system followed in the study centre. 

5. Provide more information in the concluding message to demonstrate the significance of the study.

6. Correct the typographical error's at line no. 359,382 (Age and Ageing) and 430.

Round 2
